# Cancer-testis Antigen FATE1 Expression in Adrenocortical Tumors Is Associated with A Pervasive Autoimmune Response and Is A Marker of Malignancy in Adult, but Not Children, ACC

**DOI:** 10.3390/cancers12030689

**Published:** 2020-03-14

**Authors:** Mabrouka Doghman-Bouguerra, Pascal Finetti, Nelly Durand, Ivy Zortéa S. Parise, Silviu Sbiera, Giulia Cantini, Letizia Canu, Ségolène Hescot, Mirna M. O. Figueiredo, Heloisa Komechen, Iuliu Sbiera, Gabriella Nesi, Angelo Paci, Abir Al Ghuzlan, Daniel Birnbaum, Eric Baudin, Michaela Luconi, Martin Fassnacht, Bonald C. Figueiredo, François Bertucci, Enzo Lalli

**Affiliations:** 1Institut de Pharmacologie Moléculaire et Cellulaire, Université Côte d’Azur, CNRS, 660 route des Lucioles-Sophia Antipolis, 06560 Valbonne, France; doghman@ipmc.cnrs.fr (M.D.-B.); durand@ipmc.cnrs.fr (N.D.); 2NEOGENEX-CANCER CNRS International Associated Laboratory, 660 route des Lucioles, Sophia Antipolis, 06560 Valbonne, France; 1532 Av. Silva Jardim, Curitiba PR 80250-200, Brazil; ivyparise@gmail.com (I.Z.S.P.); mirnafigueiredo@hotmail.com (M.M.O.F.); heloisakomechen@gmail.com (H.K.); bonaldf@yahoo.com.br (B.C.F.); 3Laboratoire d’Oncologie Prédictive, CRCM, Institut Paoli-Calmettes, INSERM UMR1068, CNRS UMR7258, Aix-Marseille Université, 232 Bd. Ste-Marguerite, 13009 Marseille, France; FINETTIP@ipc.unicancer.fr (P.F.); daniel.birnbaum@inserm.fr (D.B.); BERTUCCIF@ipc.unicancer.fr (F.B.); 4Department, Pelé Pequeno Principe Research Institute, 1532 Av. Silva Jardim, Curitiba PR 80250-200, Brazil; 5Division of Endocrinology and Diabetes, Department of Internal Medicine I, University Hospital, University of Würzburg, 2 Josef-Schneider-Straße, 97080 Würzburg, Germany; Sbiera_S@ukw.de (S.S.); e_sbiera_i@ukw.de (I.S.); Fassnacht_M@ukw.de (M.F.); 6Comprehensive Cancer Center Mainfranken, University of Würzburg, 6 Josef-Schneider-Straße, 97080 Würzburg, Germany; 7Endocrinology Unit, Department of Experimental and Clinical Biomedical Sciences “Mario Serio”, University of Florence, 6 viale Pieraccini, 50139 Florence, Italy; giulia.cantini@unifi.it (G.C.); letizia.canu@unifi.it (L.C.); michaela.luconi@unifi.it (M.L.); 8Service de Médecine Nucléaire, Institut Curie, 35 rue Dailly, 92210 Saint Cloud, France; segolene.hescot@curie.fr; 9Division of Pathological Anatomy, Department of Health Sciences, University of Florence, 6 viale Pieraccini, 50139 Florence, Italy; gabriella.nesi@unifi.it; 10Department of Neuro-Endocrine Tumors, Institut Gustave Roussy, 114 rue Edouard Vaillant, 94800 Villejuif, France; Angelo.paci@gustaveroussy.fr (A.P.); Abir.alghuzlan@gustaveroussy.fr (A.A.G.); Eric.BAUDIN@gustaveroussy.fr (E.B.); 11Inserm, Institut de Pharmacologie Moléculaire et Cellulaire, 660 route des Lucioles - Sophia Antipolis, 06560 Valbonne, France

**Keywords:** adrenocortical carcinoma, cancer-testis antigens, autoantibodies, immune response

## Abstract

The SF-1 transcription factor target gene *FATE1* encodes a cancer-testis antigen that has an important role in regulating apoptosis and response to chemotherapy in adrenocortical carcinoma (ACC) cells. Autoantibodies directed against FATE1 were previously detected in patients with hepatocellular carcinoma. In this study, we investigated the prevalence of circulating anti-FATE1 antibodies in pediatric and adult patients with adrenocortical tumors using three different methods (immunofluorescence, ELISA and Western blot). Our results show that a pervasive anti-FATE1 immune response is present in those patients. Furthermore, *FATE1* expression is a robust prognostic indicator in adult patients with ACC and is associated with increased steroidogenic and decreased immune response gene expression. These data can open perspectives for novel strategies in ACC immunotherapy.

## 1. Introduction

Adrenocortical carcinoma (ACC) is a rare but aggressive endocrine malignancy, which often has a somber prognosis especially when diagnosed at an advanced stage [1]. Recent studies spurred by significant technological advances in the genomic and molecular classification of ACC have given important contributions to identify molecular actors involved in ACC aggressiveness and markers of malignancy [2,3,4]. Among the determinants of malignant behavior of this tumor, a critical role is played by molecules modulating cell death and resistance to chemotherapeutic agents. We have shown that the Steroidogenic Factor-1 (SF-1) target gene *FATE1* encodes a protein localized at the interface between mitochondria and ER, which regulates Ca^2+^-dependent and mitotane-induced apoptosis in ACC cells by modulating the distance between the two organelles. FATE1 is expressed at high levels in about 40% of adult ACC and its expression is significantly and inversely correlated with patients’ overall survival (OS) [5]. Additionally, *FATE1* silencing increased sensitivity of the NCI-H1155 non-small lung cancer cell line to paclitaxel [6] and reduced viability of a variety of other cancer cell lines [7,8]. 

FATE1 belongs to the group of cancer-testis antigens, proteins whose expression is restricted to the gonads in the physiological setting, while it is reactivated in several tumor types. Importantly, an immune response against those proteins is detectable in cancer patients, probably because they are detected as “non-self” if expressed outside an immunoprivileged organ such as the testis [9]. Because of their cancer-restricted expression profiles and their immunogenic properties, cancer-testis antigens are promising targets for tumor immunotherapy [10]. In the case of FATE1, circulating antibodies directed against this protein were detected in 3/41 (7.3%) [11] and 4/52 (7.7%) [12] in two different studies in patients with hepatocellular carcinoma. Only patients expressing *FATE1* mRNA in the tumor had circulating anti-FATE1 antibodies detectable using ELISA or Western blot. 

Our study aimed to investigate the prevalence of circulating antibodies present in patients with both benign and malignant adrenocortical tumors (ACT) using three different methods (immunofluorescence (IF), ELISA and Western blot (WB)) and to identify transcripts significantly associated with low and high *FATE1* expression in ACC. Our data reveal the presence of a pervasive anti-FATE1 immune response in ACT, confirm and extend the prognostic value of *FATE1* expression in ACC and highlight a robust set of directly and inversely *FATE1*-coregulated genes in ACC gene expression profiles, which are enriched in transcripts involved in steroidogenesis and immune response, respectively. 

## 2. Results

### 2.1. FATE1 Expression and Circulating Anti-FATE1 Antibodies in Children with ACC

Using immunohistochemistry (IHC), we investigated FATE1 expression in a series of 27 children ACT (all positive for the TP53 R337H mutation). FATE1 was detectable in 26/27 tumors and its expression was relatively elevated (H-score >1) in 6/27 (22%) cases (Table 1). FATE1 expression was in most cases inhomogeneous in the tumors, with groups of FATE1-positive cells being intermingled with FATE1-negative tumor tissue (Figure 1A). The FATE1 H-score was not significantly correlated with any clinical parameter (sex, age at diagnosis, tumor stage, tumor weight, hormonal status, treatment), nor with patients’ disease-free survival (DFS) (Figure 1B). Furthermore, *FATE1* mRNA expression levels were significantly higher in a different cohort of 28 pediatric ACC cases (eight with WT TP53, 10 bearing the R337H and 10 other TP53 mutations; Table 1) [13] than in age-matched normal adrenal glands (Figure 1C). There was no significant correlation between *FATE1* mRNA expression levels and patients’ DFS (Figure 1D).

We also measured the prevalence of circulating anti-FATE1 autoantibodies in a cohort of 12 children with ACT (all positive for TP53 R337H). For 10 of them, FATE1 IHC expression data in their tumors were available, expressed in all tumors with H-scores ranging from to 0.1 to 3 (Table 1). Circulating anti-FATE1 antibodies were detected in 6/12 (50%), 2/12 (17%) and 12/12 (100%) patients’ sera using IF, ELISA and WB, respectively (Table 1). In total, 6/12 (50%) sera were positive using two (IF or ELISA in addition to WB) out of three methods and only one serum (8%) was found positive using all three methods. There was no correlation between the presence of circulating anti-FATE1 antibodies, stage of the disease, steroid secretion pattern or prognosis. 

### 2.2. FATE1 Expression and Circulating Anti-FATE1 Antibodies in Adult ACC 

*FATE1* mRNA expression is preponderant in ACC among all malignancies in the TCGA pan-cancer dataset (Figure 2A). These data are consistent with our previous results, which showed only minimal or undetectable FATE1 protein expression by IHC in a variety of other cancers [5]. Similar to pediatric tumors, FATE1 expression was heterogeneous in adult ACC (Figure 2B). 

We extended our mRNA analysis to 189 adult ACC samples (including the 79 TCGA samples) and 18 normal adrenal tissues. *FATE1* mRNA expression level was heterogeneous across all samples with a range of intensities over nearly five decades in log10 scale (Figure 2C). *FATE1* was significantly overexpressed in primary tumors when compared with normal samples (*p* = 4.47 × 10^−16^, Student *t*-test). A total of 98 out of 189 (52%) tumors showed high expression, defined as exceeding more than two-fold the mean expression level present in the normal adrenal tissues, and 91 (48%) showed low expression.

The 79 samples of the TCGA set were clinically annotated, allowing to search for a correlation between FATE1 expression and clinicopathological variables (Appendix A). High *FATE1* expression was significantly correlated to excess steroid hormone secretion and *MKI67* mRNA expression (Table 2). No correlation existed with patients’ age, sex, ENSAT stage, pathological tumor size, pathological lymph node status, surgical margins status, Weiss classification, adjuvant mitotane therapy, adjuvant radiotherapy and *TP53* mutation status. 

We assessed the prognostic value of *FATE1* mRNA expression for post-operative DFS and OS in the 64 patients non-metastatic at diagnosis (M0) from the TCGA ACC cohort. Regarding DFS, 57 patients remained event-free during a median follow-up of 22 months (range 1 to 154) and 27 (42%) displayed a DFS event. The five-year DFS rate was 52% (95% CI, 39–70; Figure 3A). High *FATE1* expression was an unfavorable prognostic parameter (Table 2). The five-year DFS rate was 64% (95% CI, 46–90) in the “*FATE1*-low” class versus 43% (95% CI, 27–68) in the “*FATE1*-high” class (*p* = 2.44 × 10^−2^, log-rank test; Figure 3B). In univariate analysis, pathological tumor size, surgical margins and TP53 status, *MKI67* and *FATE1* mRNA expression were significantly correlated to DFS. In multivariate analysis, only pathological tumor size and *FATE1* expression remained significant (Table 3). Similar results were observed with OS. Fifty patients remained event-free and 14 (22%) died. The five-year OS rate was 71% (95% CI, 57–88; Figure 3C). The five-year DFS rate was 81% (95% CI, 63–100) in the “*FATE1*-low” class versus 63% (95% CI, 45–88) in the “*FATE1*-high” class (*p* = 3.45 × 10^−2^, log-rank test; Figure 3D). Univariate analysis showed the same results as for DFS (Table 3). In multivariate analysis, pathological tumor size remained significant and *FATE1* expression tended towards significance (Table 3). Of note, the same independent prognostic value was observed for *FATE1* mRNA expression for both DFS and OS when analyzed as continuous value (*p* = 1.84 × 10^−3^ and *p* = 4.34 × 10^−2^, respectively, in the multivariate analysis and Wald test).

The presence of circulating antibodies directed against FATE1 was assessed in a cohort of adult patients with adrenocortical tumors from three different ENSAT centers (46 malignant—among these, six patients had a pre-surgery and a post-surgery sample available for analysis—and 10 benign). Patients’ data are reported in Table 4. Anti-FATE1 antibodies were detected in 20/62 (32%), 44/62 (71%) and 62/62 (100%) of patients’ sera using IF, ELISA and WB, respectively (Table 4). In total, 30/62 (48%) sera were positive using two (IF or ELISA in addition to WB) out of three methods and 17/62 (27%) sera were positive using all three methods. There was no correlation between the presence of circulating anti-FATE1 antibodies, stage of the disease, steroid secretion pattern or prognosis.

### 2.3. FATE1-Coregulated Genes in Adult ACC 

Next, to further explore the biological pathways associated with *FATE1* mRNA expression in ACC, we compared the expression profiles of all genes between “*FATE1*-high” tumors and “*FATE1*-low” tumors within the whole TCGA data set (learning set). We identified 1084 differentially expressed genes, including 659 genes upregulated and 425 genes downregulated in the “*FATE1*-high” samples (Figure 4A and Appendix A). The robustness of this gene signature was confirmed in the independent validation set (pool of three other public ACC gene expression datasets; *n* = 110; Figure 4B) with high significance. Ontology analysis of these 1084 differentially expressed genes revealed that expression of genes involved in steroidogenesis is associated with high *FATE1* expression, while expression of immune response genes is associated with low *FATE1* expression (Figure 4C, Appendix A). 

## 3. Discussion

ACC is considered as an immunologically “cold” cancer type. Pan-cancer genomic studies have revealed that ACC has on average one of the lowest immune signature activation scores and belongs to the “lymphocyte-depleted” cancer subtype [4,14,15]. A major determinant of this situation can be found in the well-known immunosuppressive activity of glucocorticoids produced by the tumor. Mutations in the beta-catenin and *TP53* pathways, which are frequently found in ACC, may also contribute to the impairment of anti-tumor immune response [16]. These data, together with the reported low expression of the immune checkpoint molecule PDL1 in ACC tumor cells [17], suggest that chances for successful immunotherapy are limited in patients with ACC. One study reported that a subset of platinum-pretreated patients with metastatic ACC had a partial response or stable disease after treatment with the anti-PDL1 antibody avelumab [18], but these results were similar to those obtained in other studies using second-line chemotherapy [19]. However, recent studies showed that the immune response may have an important role in influencing the clinical course of patients with ACC. In pediatric ACC, MHC class II expression by tumor-infiltrating hematopoietic cells and the number of CD8^+^ T-lymphocytes are important prognostic indicators [13,20]. Even if in general only a small percentage of ACC tumor cells express PDL1, as detected by IHC [17,18], higher levels of *PDL1* mRNA expression are significantly correlated with an inflammatory gene expression signature and longer DFS of adult patients with ACC [21]. Furthermore, an unpublished study from the Würzburg group has shown that the number of tumor-infiltrating CD4^+^ and CD8^+^ T-lymphocytes are stage-independent prognostic indicators in patients with ACC and that glucocorticoid excess is associated with T-cell depletion and unfavorable prognosis [22]. These data are consistent with previous classifications based on genomic profiles, which showed that steroid phenotype-high ACC, especially those with higher proliferation scores, have the worst prognosis [4].

Our results show that patients with adrenocortical tumors (both benign and malignant) can mount an immune response against FATE1, as shown by the widespread presence of circulating antibodies directed against this cancer-testis antigen. High tumor *FATE1* mRNA expression levels in adult ACC are associated with high steroidogenic gene expression, immune cell depletion and worse prognosis. Conversely, *FATE1* expression is not prognostically relevant in pediatric ACC, which has a distinct biological and clinical profile compared with tumors in adults [23]. An explanation for the coregulation of *FATE1* and steroidogenic genes is probably found in their common transcriptional regulator SF-1 [24,25]. In pediatric ACT, SF-1 expression is increased in the majority of cases compared to normal adrenal, without prognostic significance [26,27], while in adult ACC, SF-1 is overexpressed in a subset of cases and is a robust marker of malignancy [28]. High steroid production by *FATE1*-expressing tumors is likely to create an unfavorable *milieu* for immune cell infiltration and local response against this antigen. On the other side, FATE1 expression in the most aggressive group of ACC could open new perspectives for immunotherapy using vaccination against this and other cancer neoantigens [10], possibly in combination with steroidogenic inhibitors to counteract the immunosuppressive action of glucocorticoids. 

## 4. Materials and Methods

### 4.1. Biological Materials

All adult patients or the guardians for pediatric patients provided written informed consent for collecting blood, tissue and clinical data, including follow-up and survival data. Sera were stored at −20 °C until analysis and tumor samples were paraffin-embedded or preserved in liquid nitrogen before RNA extraction. Serum, RNA and tissue samples were analyzed anonymously, following the principles of the Declaration of Helsinki and the Good Clinical Practice Guidelines. The study was approved by the Institutional Review Boards of the Pequeno Principe Hospital (CAAE 0612.0.015.000-08), University of Würzburg (#88/11), University of Florence (CEAVC Em. 2019-201-26/11/2019) and Institut Gustave Roussy (PP 13-021). 

### 4.2. Immunohistochemistry on Paraffin-Embedded Tissue

Tissue sections from pediatric (*n* = 27) and adult (*n* = 5) ACC were deparaffinized with xylene and rehydrated with a graded alcohol series. Epitope retrieval was carried out in 10 mM citrate buffer (pH 6.0) for 20 min at 100 °C in a low-pressure cooker, followed by cooling for 20 min at room temperature. After a permeabilization step and inactivation of endogenous peroxidase with REAL Peroxidase-Blocking Solution (Dako, Santa Clara, CA, USA), sections were blocked with 10% normal goat serum for 1 h at room temperature and then incubated with the anti-FATE1 antibody (clone 6A11; Abcam, Cambridge, UK) overnight at 4 °C. After washing, HRP-conjugated anti-mouse IgG (EnVision+ System-HRP, Dako) was added and further incubated for 30 min at room temperature. The slides were washed and diaminobenzidine tetrahydrochloride was then added (Liquid DAB+ substrate chromogen system, Dako), followed by counterstaining with hematoxylin solution. Sections of normal human testis were used as positive control. All slides were analyzed independently by two investigators (M.D.B. and E.L.) blinded as to the samples’ clinical data. Staining intensity was graded as negative (0), low to medium (1) or strong (2). The percentage of positive tumor cells was calculated for each specimen and scored 0 if 0% were positive, 0.1 if 1–9% were positive, 0.5 if 10–49% were positive and 1 if 50% or more were positive. A semiquantitative H-score was calculated by multiplying the staining intensity grading score by the proportion score, as described [5]. 

### 4.3. Gene Expression Data in Pediatric Patients with ACC

They were retrieved by NCBI GEO, accession code GSE76019 [13].

### 4.4. Cell Culture and Indirect Immunofluorescence Assay for the Detection of Anti-FATE1 Antibodies in Patients’ Sera

The H295R/TR N-Flag FATE1 stable cell line was established and cultured as described [5] and can express N-terminally flagged FATE1 in a doxycycline-inducible manner. To induce *FATE1* overexpression, cells were treated for 24 h with doxycyline (Dox; 1 g/mL, Sigma-Aldrich, St. Quentin Fallavier, France). We developed an IF assay to measure the presence of anti-FATE1 antibodies in patients’ sera by exploiting our H295R/TR Nflag-FATE1 cell line. Briefly, cells were seeded on 96-well Clear Bottom Black Polystyrene microplates (Corning, Corning, NY, USA) and treated for 24 h without or with doxycycline. Immunofluorescence was performed as previously described [5]. Cells were incubated overnight at 4 °C with different dilutions of patients’ sera (1/20 to 1/100) prepared in phosphate-buffered saline (PBS/0.5%) bovine serum albumin (BSA) and then with an Alexa Fluor 488-conjugated secondary antibody (Invitrogen, Carlsbad CA, USA). Cell nuclei were counterstained with 4,6-diamidino-2-phenylindole (DAPI). Images were collected with the Cytation 5 Cell Imaging Multi-Mode Reader (BioTek, Colmar, France) with combined cell imaging and analysis of mitochondrial staining pattern in Dox-treated H295R/TR FATE1 cells compared with non-treated cells. An example of an ACC serum positive for anti-FATE1 antibodies detectable in IF is shown in Figure 5A. 

### 4.5. Expression and Purification of FATE 1–162

The cDNA encoding the extracellular domain of the FATE1 protein (aa. 1–162) was inserted into the pET-15b bacterial expression plasmid. Production and purification of the FATE1 1–162 protein was performed by the Structural Biology and Genomics Technology Platform (CBI-IGBMC, Illkirch, France). Briefly, pET-FATE1 1–162 was transformed into BL21 Star (DE3) *Escherichia coli*. After induction by 1 mM IPTG at 37 °C overnight, the purification of recombinant proteins was performed by Ni^2+^ affinity chromatography. Fractions were desalted by gel filtration chromatography using a S75 16/500 column, and further purified on a UnoS1 cation-exchange column eluted with a linear gradient of 0–1 M NaCl in 20 mM Hepes pH 8, 2 M urea buffer. 

### 4.6. ELISA for the Detection of Anti-FATE1 Antibodies in Patients’ Sera

We developed an ELISA assay for the detection of autoantibodies against FATE1 in patients’ sera, as described previously for other autoantigens [29,30]. To exclude serum sample-specific background noise caused by unspecific hydrophobic binding of immunoglobulins to the solid surface and/or by protein-protein interaction, plates were coated in parallel with the same amount of BSA per well. Briefly, recombinant human FATE1 1–162 and BSA (BSA fraction V; Roche) were coated to ELISA microplates (Maxisorp, Nunc, Roskilde, Denmark) in 20 mM Tris pH 8.0 (100 μL/well, 1 μg/mL) overnight at 4 °C. Plates were washed with PBS/0.05% Tween-20 and blocked for 2 h with SeramunBlock (Seramun Diagnostica, Heidesee, Germany). We analyzed sera from patients with benign adrenal tumors (ACA) and adult ACC that were provided by three ENSAT centers (Würzburg, Germany, *n* = 28; Florence, Italy *n* = 22; Villejuif, France, *n* = 12). In addition, sera from patients with pediatric ACC were analyzed (*n* = 12; Curitiba, PR, Brazil). The diagnosis of ACC was made on established criteria based on clinical, biochemical, and morphological data and clinical data were collected through ENSAT and hospital centers registries. Patients’ sera were diluted at 1:100 in PBS/0.1% non-fat dry (NFD) milk and added in duplicate to the ELISA plates (100 μL per well). After 2 h incubation at room temperature on a plate shaker, plates were washed five times with PBS/0.05% Tween-20. Anti-human IgG-horseradish peroxidase (HRP) conjugate (Southern Biotech #2010-05) diluted 1:20,000 in PBS/0.1% NFD was added (100 μL per well) and incubated for 1 h at room temperature on a plate shaker. After five washes, tetramethylbenzidine (TMB) was added, and the reactions were developed and then stopped with stop solution (TMB solution for ELISA; Interchim, Montluçon, France). Plates were read at 450 nm. A positive serum was used in each plate as a positive control. We defined a positive reaction as an OD value of a diluted serum that exceeds the mean OD value of negative control (coated with irrelevant protein) by two-fold. Results for sera from ACC and ACA patients tested in ELISA are shown in Figure 5B. 

### 4.7. Western Blot Analysis for the Detection of Anti-FATE1 Antibodies in Patients’ Sera

SDS-PAGE of FATE1 1–162 followed by multiscreen immunoblotting was used to determine the presence of anti-FATE1 antibodies in patients’ sera. Briefly, 1 μg of FATE1 1–162 recombinant protein and BSA as a negative control were separated on a 12% SDS–polyacrylamide gel electrophoresis and transferred onto nitrocellulose membranes. After a blocking step with 5% NFD milk in PBS/0.1% Tween-20 (PBS-T), the membranes were incubated with patients’ sera diluted 1:100 in 0.5% NFD milk in PBS-T overnight at 4 °C using the Mini-Protean II Multiscreen apparatus (Bio-Rad). The secondary antibody was HRP-conjugated goat anti-human IgG at 1:10,000 dilution. The detection of protein bands was performed with an enhanced chemiluminescent substrate (ECL Prime, GE Healthcare, Chicago, IL, USA) and a LAS3000 digital imager (Fuji, Saint-Quentin-en-Yvelines, France). Band intensities were quantified using the ImageJ software. Examples of sera from ACC patients tested in Western blot and specificity controls are shown in Figure 5C,D.

### 4.8. Gene Expression Analysis in Adult ACC Cohorts

We gathered clinicopathological and mRNA expression data of clinical ACC samples from four publicly available data sets [4,32,33,34], comprising at least one probe set representing *FATE1* (Appendix A). All samples were pre-treatment primary tumor samples collected on the surgical resection specimen. The TCGA set [4] included 79 cases profiled using RNA-seq (Illumina, San Diego, CA, USA) and clinically annotated allowing prognostic analysis (Appendix A). The three other data sets (110 ACC and 18 normal adrenal samples) were not annotated for both expression and prognostic/survival data. Data analysis required pre-analytic processing, as previously described [21]. To explore more-in-depth the biological pathways associated with *FATE1* mRNA expression in ACC, we applied a supervised analysis to the whole TCGA data set as a learning set (*n* = 79) and compared the expression profiles of all genes between tumors with low versus high *FATE1* mRNA expression using a moderated *t*-test with the following significance thresholds: *p* < 0.05, q < 0.25 and fold change (FC) higher than |2×|. Ontology analysis applied to the resulting gene list was based on GO biological processes of the Database for Annotation, Visualization and Integrated Discovery (DAVID; david.abcc.ncifcrf.gov/). The robustness of this gene list was tested in the 110 remaining ACC samples used as an independent validation set. We computed for each sample a “*FATE1* metagene” score as the difference between the mean expression of genes upregulated and the mean expression of genes downregulated in the “*FATE1*-high” class. The natural score of 0 was used as threshold to define a sample as “*FATE1*-high” (positive score) or “*FATE1*-low” (negative score). Once defined, we analyzed the concordance of the predicted and observed statutes of all 110 samples using the Student *t*-test. Correlations between tumor classes and clinicopathological features were analyzed using the *t*-test or Fisher’s exact test when appropriate. DFS was calculated from the date of diagnosis until the date of distant relapse or death from any cause. OS was calculated from the date of diagnosis until the date of death from any cause. Follow-up was measured from the date of diagnosis to the date of last news for event-free patients. Survivals were calculated using Kaplan-Meier method and curves were compared with log-rank test. Univariate and multivariate survival analyses were done using Cox regression analysis (Wald test). Variables tested in univariate analyses included patients’ age at time of diagnosis, sex, ENSAT stage (1 vs. 2–3), pathological tumor size (pT: pT1 vs. pT2, vs. pT3, vs. pT4), pathological lymph node status (pN: negative vs. positive), surgical margins status (R0 vs. R1, vs. R2), Weiss classification (low vs. high), excess of steroid hormonal secretion (no vs. yes), adjuvant mitotane therapy (no vs. yes), adjuvant radiotherapy (no vs. yes), MKI67 mRNA expression (low vs. high), and TP53 mutation status (wild-type vs. mutated). Variables with a *p*-value <0.05 in univariate analysis were tested in multivariate analysis. All statistical tests were two-sided at the 5% level of significance. Statistical analysis was done using the survival package (version 2.30) in the R software (version 2.15.2; http://www.cran.r-project.org/). We followed the reporting REcommendations for tumor MARKer prognostic studies (REMARK criteria) [35].

## 5. Conclusions

We have shown that a pervasive circulating autoantibody response against the cancer-testis antigen FATE1 is present in pediatric and adult patients with ACC. High *FATE1* expression is negatively correlated to DFS and OS in adult patients with ACC and associated with increased steroidogenic and decreased immune response gene expression. These data can open new perspectives for ACC immunotherapy. 

## Figures and Tables

**Figure 1 cancers-12-00689-f001:**
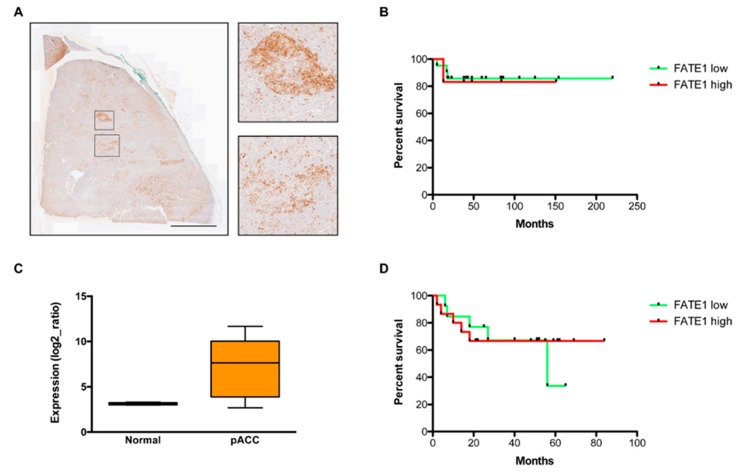
FATE1 expression and correlation with prognosis in pediatric ACC. (**A**) FATE1 IHC staining in a pediatric ACC. Two groups of FATE1-positive cells are shown at higher magnification. Scale bar, 5 mm. (**B**) Disease-free survival analysis in a cohort of children with ACC (*n* = 27) according to their low (H-score ≤1; 21 patients) and high (H-score >1; 6 patients) tumor FATE1 expression. Log-rank, *p* = 0.8660. (**C**) *FATE1* mRNA expression in normal children adrenal (*n* = 5) and in a cohort of pediatric ACC (pACC; *n* = 28). *p* = 0.0072, Mann-Whitney test. (**D**) Disease-free survival analysis of children with ACC according to their low (<2× expression compared to normal adrenal; 13 patients) and high (>2× expression compared to normal adrenal; 15 patients) tumor *FATE1* mRNA expression. Log-rank, *p* = 0.8212.

**Figure 2 cancers-12-00689-f002:**
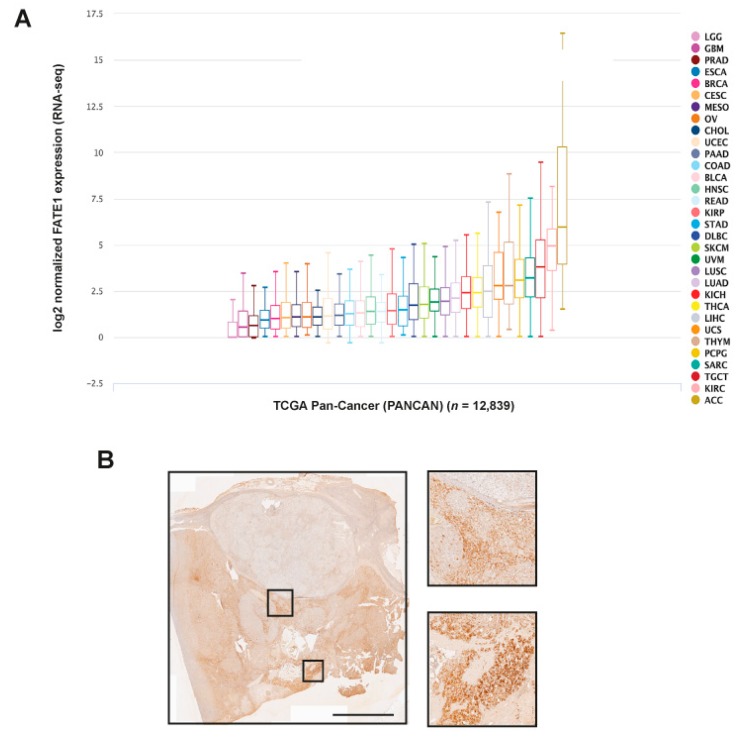
FATE1 expression in adult ACC. (**A**) *FATE1* mRNA expression in cancers of the TCGA PANCAN dataset. 12,839 cases in total were analyzed using the Xena browser (https://xenabrowser.net). Tumor types are color-coded. ACC, adrenocortical carcinoma; KIRC, kidney clear cell carcinoma; TGCT, testicular germ cell cancer; SARC, sarcoma; PCPG, pheocromocytoma-paraganglioma; THYM, thymoma; UCS, uterine carcinosarcoma; LIHC, liver hepatocellular carcinoma; THCA, thyroid carcinoma; KICH, kidney chromophobe cell carcinoma; LUAD, lung adenocarcinoma; LUSC, lung squamous cell carcinoma; UVM, uveal melanoma; SKCM, skin cutaneous melanoma; DLBC, diffuse large B-cell lymphoma; STAD, stomach adenocarcinoma; KIRP, kidney papillary cell carcinoma; READ, rectum adenocarcinoma; HNSC, head and neck squamous cell carcinoma; BLCA, bladder urothelial carcinoma; COAD, colon adenocarcinoma; PAAD, pancreatic adenocarcinoma; UCEC, uterine corpus endometrial carcinoma; CHOL, cholangiocarcinoma; OV, ovarian serous cystadenocarcinoma; MESO, mesothelioma; CESC, cervical squamous cell carcinoma and endocervical adenocarcinoma; BRCA, breast invasive carcinoma; ESCA, esophageal carcinoma; PRAD, prostate adenocarcinoma; GBM, glioblastoma multiforme; LCG, brain lower grade glioma. (**B**) FATE1 IHC staining in an ACC from an adult patient. Some tumor areas display intense FATE1 staining, other nodules appear to express very little FATE1. Two groups of FATE1 positive cells are shown at higher magnification. Scale bar, 5 mm. (**C**) *FATE1* mRNA expression across 189 adult ACC and 18 normal adrenal samples. Histogram of distribution of FATE1 mRNA expression levels. The red horizontal line defines the “*FATE1*-low” and “*FATE1*-high” classes.

**Figure 3 cancers-12-00689-f003:**
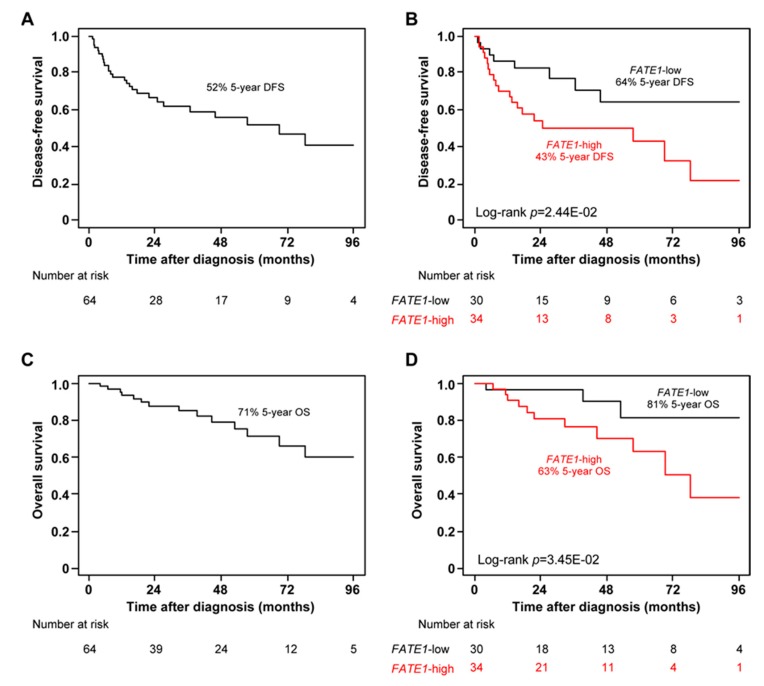
High *FATE1* mRNA expression is an unfavorable prognostic variable in adult M0 ACC from TGCA. (**A**) Kaplan-Meier DFS curves in the whole population (*n* = 64). (**B**) Similar to A, but in the “*FATE1*-low” and “*FATE1*-high” classes. (**C**) Similar to A, but for OS. (**D**) Similar to B, but for OS.

**Figure 4 cancers-12-00689-f004:**
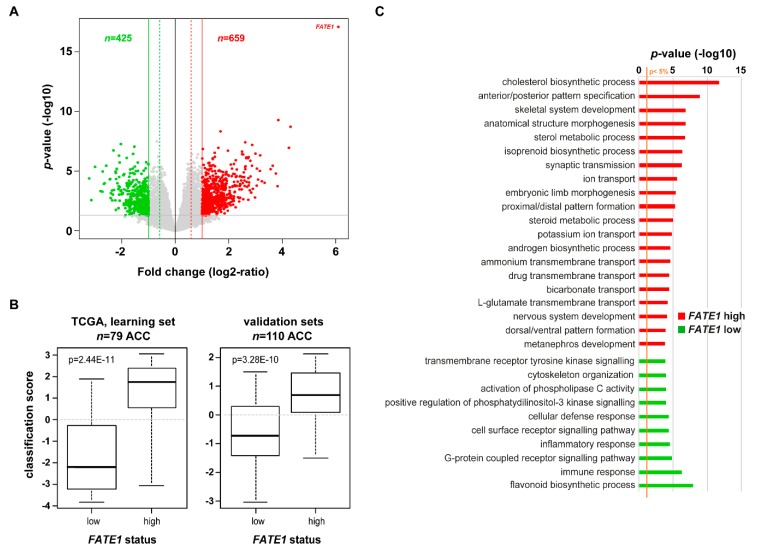
*FATE1*-coregulated genes in ACC. (**A**) Volcano plot of transcripts were significantly associated with *FATE1* high- (*n* = 44) and low-expressing (*n* = 35) ACC within the TCGA cohort. (**B**) Validation of the 1084-gene signature: box-plot of the metagene-based prediction score in the “*FATE1*-high” samples compared to the “*FATE1*-low” samples. The *p*-value is for the Student’s t-test assessing the difference of the prediction score between the observed *FATE1* classes, which is, as expected, very significant not only in the learning set (left), but also in the validation set (right), showing the robustness of the signature (**C**) Biological Processes (GO) differentially coregulated in association with *FATE1* high vs. low expression levels.

**Figure 5 cancers-12-00689-f005:**
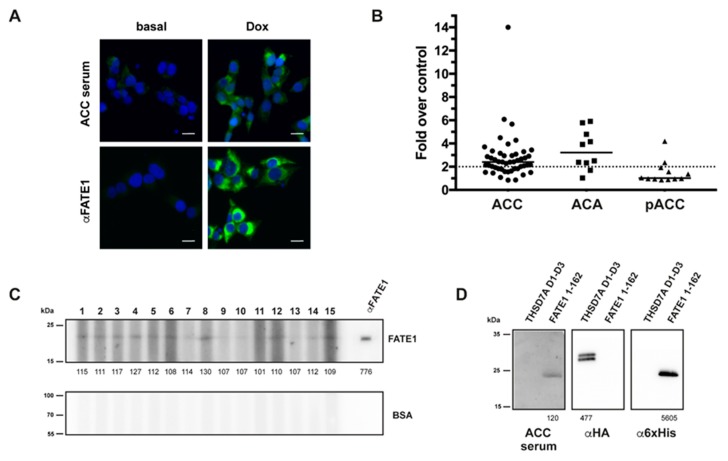
IF, ELISA and Western blots for circulating anti-FATE1 antibodies. (**A**) IF: mitochondrial staining by an ACC serum selectively in Dox-treated, but not untreated, H295R/TR N-Flag FATE1 cells. Anti-FATE1 monoclonal antibody staining is shown as a control. Scale bar, 20 μm. (**B**) ELISA: graph showing results for ACC, ACA and pediatric ACC (pACC). Samples generating OD_450_ signals higher than two-fold negative controls (dotted line) were considered as positive. (**C**) Western blot: sera reactivities were tested against recombinant FATE1 (1–162) and BSA as negative control. A representative blot is shown using samples from patients 1 to 15 in Table 4. Band signals (values under each lane in the FATE1 immunoblot) are expressed as a percentage over the local background. (**D**) Specificity control for Western blot: a serum from a patient with ACC recognizes recombinant FATE1 (aa. 1–162) but not another recombinant human autoantigen of similar molecular weight (thrombospondin type-1 domains 1 to 3 of THSD7A [31]). Recombinant proteins were detected with anti-hemagglutinin (HA; THSD7A) and anti-6xHis-Tag (FATE1) antibodies, respectively. Band signals (values under each lane) are expressed as a percentage over the local background. Original uncropped blots are shown in Appendix A.

**Table 1 cancers-12-00689-t001:** Clinical, histopathological, serological and gene expression features of children with ACC in this study.

Patient	Gender	TP53	Age at Diagnosis (Years)	Staging	Clinical Manifestations	Weight (g)	Histopathology	Treatment	Recurrence	Outcome	DFS (Months)	FATE1 H-score	Anti-FATE1 Abs IF+	Anti-FATE1 Abs ELISA+	Anti-FATE1 Abs WB+	*FATE1* mRNA Expression Data
1	F	R337H	10	1	V	110	ACC	S + C	Yes	DD	17	1	+	-	+	NA
2	M	R337H	7	2	V	238	ACC	S + C	No	Alive	65	0.1	+	-	+	NA
3	F	R337H	7	4	V	342	ACC	S + C	Yes	DD	18	1	+	+	+	NA
4	F	R337H	1	1	V	16	ACC	S + C	No	Alive	38	3	-	-	+	NA
5	M	R337H	1	2	CS + HBP	126	ACC	S	No	Alive	43	0.5	+	-	+	NA
6	F	R337H	5	2	V	318	ACC	S + C	No	Alive	48	1	-	-	+	NA
7	M	R337H	0	1	NF	50	ACC	S	No	Alive	41	0.2	+	-	+	NA
8	F	R337H	1	1	V + CS	62	ACC	S	No	Alive	38	1.5	-	-	+	NA
9	F	R337H	1	2	V	253	ACC	S	No	Alive	19	0.1	+	-	+	NA
10	M	R337H	1	2	V + CS	105	ACC	S	No	Alive	38	1	-	-	+	NA
11	F	R337H	7	4	V	184	ACC	S + C	No	DD	16	NA	-	+	+	NA
12	F	R337H	1	3	V	32	ACC	S + C	Yes	DD	25	NA	-	-	+	NA
13	M	R337H	1	1	V + CS	12	ACC	S	No	Alive	84	3	NA	NA	NA	NA
14	F	R337H	1	2	V + CS	212	ACC	S	No	Alive	83	0.1	NA	NA	NA	NA
15	F	R337H	3	3	V	125	ACC	S + C	No	DD	5	1	NA	NA	NA	NA
16	M	R337H	10	1	V + CS	NA	ACC	S + C	No	Alive	23	0.2	NA	NA	NA	NA
17	F	R337H	2	2	V	275	ACC	S	No	Alive	38	3	NA	NA	NA	NA
18	M	R337H	3	2	V	300	ACC	S	No	Alive	151	2	NA	NA	NA	NA
19	F	WT	9	1	V + CS	15	ACC	S	No	Alive	86	1	NA	NA	NA	NA
20	M	R337H	2	4	V + CS	80	ACC	S + C	No	Alive	154	1	NA	NA	NA	NA
21	F	R337H	2	2	V	127	ACC	S	No	Alive	106	0.1	NA	NA	NA	NA
22	F	R337H	2	1	V	77	ACC	S	No	Alive	38	1	NA	NA	NA	NA
23	F	WT	2	1	V + CS	82	ACC	S	No	Alive	84	0.3	NA	NA	NA	NA
24	F	R337H	1	1	V + CS	33	ACC	S	No	Alive	48	1.5	NA	NA	NA	NA
25	F	R337H	2	4	V + CS	392	ACC	S + C	No	DD	13	1.5	NA	NA	NA	NA
26	M	R337H	1	2	NF	300	ACC	S	No	Alive	220	0.2	NA	NA	NA	NA
27	M	R337H	15	4	AbM	3150	ACC	S + C	No	Alive	48	0	NA	NA	NA	NA
28	F	WT	1	3	V + CS	98	ACC	S + C	No	Alive	60	0.1	NA	NA	NA	NA
29	F	R337H	8	3	V	300	ACC	S + C	No	Alive	125	0.2	NA	NA	NA	NA
30	F	WT	1	2	C	120	ACC	S	No	Alive	51	NA	NA	NA	NA	Yes
31	M	WT	3	3	V + CS	144	ACC	S + C	No	Alive	84	NA	NA	NA	NA	Yes
32	F	WT	1	4	V + CS	80	ACC	S + C	No	Alive	40	NA	NA	NA	NA	Yes
33	F	WT	17	4	V	1000	ACC	S + C	Yes	DD	18	NA	NA	NA	NA	Yes
34	M	WT	15	4	V + CS	579	ACC	S + C	Yes	DD	10	NA	NA	NA	NA	Yes
35	F	WT	4	1	V	69	ACC	S	No	Alive	18	NA	NA	NA	NA	Yes
36	M	WT	12	4	C	NA	ACC	C	Yes	DD	6	NA	NA	NA	NA	Yes
37	F	WT	2	2	V	190	ACC	S	No	Alive	52	NA	NA	NA	NA	Yes
38	F	R337H	4	3	V	260	ACC	S + C	No	Alive	61	NA	NA	NA	NA	Yes
39	F	R337H	3	1	V	6	ACC	S	No	Alive	62	NA	NA	NA	NA	Yes
40	F	R337H	1	3	V + CS	86	ACC	S + C	No	Alive	48	NA	NA	NA	NA	Yes
41	F	R337H	3	1	V	30	ACC	S	No	Alive	51	NA	NA	NA	NA	Yes
42	F	R337H	1	3	V	68	ACC	S + C	No	Alive	55	NA	NA	NA	NA	Yes
43	F	R337H	1	2	V	142	ACC	S	No	Alive	52	NA	NA	NA	NA	Yes
44	F	R337H	3	1	V	12	ACC	S	No	Alive	65	NA	NA	NA	NA	Yes
45	F	R337H	1	1	V	80	ACC	S	No	Alive	22	NA	NA	NA	NA	Yes
46	F	R337H	8	1	V	22	ACC	S	No	Alive	21	NA	NA	NA	NA	Yes
47	M	R337H	16	4	NF	NA	ACC	C	Yes	DD	7	NA	NA	NA	NA	Yes
48	F	splice	1	3	V	117	ACC	S + C	No	Alive	69	NA	NA	NA	NA	Yes
49	M	splice	6	2	V	810	ACC	S	Yes	DD	4	NA	NA	NA	NA	Yes
50	M	DBD	1	1	V	10	ACC	S	No	Alive	59	NA	NA	NA	NA	Yes
51	M	DBD	1	1	V	56	ACC	S	No	Alive	52	NA	NA	NA	NA	Yes
52	F	DBD	3	3	C	466	ACC	S + C	Yes	Alive	18	NA	NA	NA	NA	Yes
53	M	DBD	13	4	NF	579	ACC	S + C	Yes	DD	2	NA	NA	NA	NA	Yes
54	F	DBD	4	2	NF	137	ACC	S	Yes	DD	56	NA	NA	NA	NA	Yes
55	M	DBD	10	3	NF	627	ACC	S + C	Yes	Alive	14	NA	NA	NA	NA	Yes
56	M	DBD	5	3	NF	595	ACC	S + C	Yes	Alive	27	NA	NA	NA	NA	Yes
57	F	DBD	5	2	V	192	ACC	S	No	Alive	25	NA	NA	NA	NA	Yes

Clinical manifestations: V, virilizing tumor; CS, Cushing’s syndrome; HBP, hypertension; NF, non-functioning; AbM, abdominal mass. Treatment: S, surgery; C, chemotherapy. Outcome: DD, dead of disease. DFS: disease-free survival. NA: not available.

**Table 2 cancers-12-00689-t002:** Correlations of FATE1 mRNA expression with clinicopathological variables in adult ACC patients from TCGA.

Characteristics		*n*	*FATE1*		*p*-Value
			Low (*n* = 35)	High (*n* = 44)	
Age, median	years (range)	79	46 (17–75)	53 (14–77)	0.560
Sex					0.645
	female	48	20 (57%)	28 (64%)	
	male	31	15 (43%)	16 (36%)	
ENSAT stage					0.484
	1	9	3 (9%)	6 (14%)	
	2	37	20 (57%)	17 (40%)	
	3	16	7 (20%)	9 (21%)	
	4	15	5 (14%)	10 (24%)	
Pathological tumor size (pT)					0.487
	pT1	9	3 (9%)	6 (14%)	
	pT2	42	22 (63%)	20 (48%)	
	pT3	8	4 (11%)	4 (10%)	
	pT4	18	6 (17%)	12 (29%)	
Pathological lymph node (pN)					0.170
	pN0	68	33 (94%)	35 (83%)	
	pN1	9	2 (6%)	7 (17%)	
Surgical margins status					0.776
	R0	55	25 (81%)	30 (77%)	
	R1–R2	15	6 (19%)	9 (23%)	
Weiss classification					1.000
	low (1–3)	14	6 (23%)	8 (21%)	
	high (4–9)	50	20 (77%)	30 (79%)	
Steroid hormone secretion excess					1.32 × 10^−3^
	no	26	18 (56%)	8 (19%)	
	yes	48	14 (44%)	34 (81%)	
Adjuvant chemotherapy					0.051
	no	25	15 (45%)	10 (23%)	
	yes	51	18 (55%)	33 (77%)	
Adjuvant mitotane					0.220
	no	26	14 (44%)	12 (28%)	
	yes	49	18 (56%)	31 (72%)	
Adjuvant radiotherapy					1.000
	no	59	26 (79%)	33 (79%)	
	yes	16	7 (21%)	9 (21%)	
*TP53* mutation status					0.156
	mut	15	4 (11%)	11 (25%)	
	WT	64	31 (89%)	33 (75%)	
*MKI67* mRNA status					2.36 × 10^−2^
	low	40	23 (66%)	17 (39%)	
	high	39	12 (34%)	27 (61%)	
DFS event *			8 (27%)	19 (56%)	2.37 × 10^−2^
Five-year DFS *			64% (46–90)	43% (27–68)	2.44 × 10^−2^
OS event *			3 (10%)	11 (32%)	3.78 × 10^−2^
Five-year OS *			81% (63–100)	63% (45–88)	3.45 × 10^−2^

* M0 patients only (*n* = 64).

**Table 3 cancers-12-00689-t003:** Univariate and multivariate prognostic analyses for DFS and OS in adult M0 ACC patients from TCGA.

**DFS ***		**Univariate**			**Multivariate ****			**Multivariate ****		
		***n***	**HR [95% CI]**	***p*-Value**	***n***	**HR [95% CI]**	***p*-Value**	***n***	**HR [95% CI]**	***p*-Value**
Age, median		64	0.99 [0.97–1.02]	0.682						
Sex	male vs. female	64	0.79 [0.36–1.72]	0.546						
ENSAT stage	2–3 vs. 1	62	2.86 [0.67–12.2]	0.155						
Pathological tumor size (pT)	pT2 vs. pT1	62	2.11 [0.48–9.30]	1.01 × 10^−2^	56	2.41 [0.53–11.02]	0.257	56	3.98 [0.82–19.29]	0.087
	pT3 vs. pT1		6.99 [1.33–36.9]		56	9.39 [1.41–62.54]	2.07 × 10^−2^	56	21.1 [2.73–163.4]	3.47 × 10^−3^
	pT4 vs. pT1		9.02 [1.58–51.6]		56	20.7 [1.62–264.7]	1.98 × 10^−2^	56	28.0 [2.60–301.4]	5.98 × 10^−3^
Pathological lymph node (pN)	pN1 vs. pN0	62	1.81 [0.42–7.78]	0.423						
Surgical margins status	R1-R2 vs. R0	56	5.10 [1.13–23.0]	3.38 × 10^−2^	56	1.04 [0.13–8.59]	0.970	56	0.55 [0.08–3.69]	0.540
Weiss classification	high vs. low	49	3.00 [0.94–9.52]	0.063						
Steroid hormone secretion excess	yes vs. no	59	2.50 [0.93–6.71]	0.068						
Adjuvant mitotane	yes vs. no	61	1.44 [0.65–3.19]	0.372						
Adjuvant radiotherapy	yes vs. no	62	1.67 [0.66–4.21]	0.282						
*MKI67* mRNA status	high vs. low	64	3.60 [1.64–7.89]	1.38 × 10^−3^	56	1.65 [0.58–4.69]	0.350	56	1.19 [0.38–3.73]	0.760
*TP53* mutation status	wt vs. mt	64	0.28 [0.09–0.86]	2.58 × 10^−2^	56	0.84 [0.11–6.41]	0.870	56	0.72 [0.13–4.07]	0.707
*FATE1* class	high vs. low	64	2.51 [1.09–5.74]	2.98 × 10^−2^	56	5.18 [1.53–17.5]	8.16 × 10^−3^			
*FATE1*, continuous value		64	1.14 [1.04–1.25]	3.79 × 10^−3^				56	1.23 [1.08–1.40]	1.84 × 10^−3^
**OS ***		**Univariate**			**Multivariate ****			**Multivariate ****		
		***n***	**HR [95% CI]**	***p*-Value**	***n***	**HR [95% CI]**	***p*-Value**	***n***	**HR [95% CI]**	***p*-Value**
Age, median		64	1.00 [0.96–1.03]	0.834						
Sex	male vs. female	64	1.45 [0.51–4.15]	0.485						
ENSAT stage	2–3 vs. 1	62	2.97 [0.38–23.0]	0.298						
Pathological tumor size (pT)	pT2 vs. pT1	62	1.85 [0.22–15.5]	5.56 × 10^−3^	56	5.7 [0.48–67.17]	0.167	56	9.0 [0.68–118.0]	0.096
	pT3 vs. pT1		6.83 [0.57–81.8]		56	29.0 [1.20–700.7]	3.83 × 10^−2^	56	37.33 [1.46–952]	2.85 × 10^−2^
	pT4 vs. pT1		21.6 [2.08–224]		56	16.3 [0.68–393.8]	0.085	56	21.07 [1.00–443]	4.99 × 10^−2^
Pathological lymph node (pN)	pN1 vs. pN0	62	0.00 [0.00 - Inf.]	0.998						
Surgical margins status	R1-R2 vs. R0	56	35.7 [3.22–395]	3.59 × 10^−3^	56	3.47 [0.20–60.72]	0.395	56	2.84 [0.17–48.4]	0.472
Weiss classification	high vs. low	49	4.79 [0.92–24.8]	0.062						
Steroid hormone secretion excess	yes vs. no	59	3.17 [0.70–14.3]	0.134						
Adjuvant mitotane	yes vs. no	61	2.18 [0.65–7.29]	0.208						
Adjuvant radiotherapy	yes vs. no	62	1.06 [0.23–4.84]	0.938						
*MKI67* mRNA status	high vs. low	64	8.87 [2.45–32.2]	8.99 × 10^−4^	56	5.28 [0.97–28.80]	0.055	56	3.26 [0.51–20.9]	0.212
*TP53* mutation status	wt vs. mt	64	0.15 [0.04–0.60]	7.79 × 10^−3^	56	0.21 [0.02–2.08]	0.183	56	0.18 [0.02–1.49]	0.112
*FATE1* class	high vs. low	64	3.66 [1.01–13.2]	4.78 × 10^−2^	56	8.74 [0.95–80.52]	5.60 × 10^−2^			
*FATE1*, continuous value		64	1.19 [1.06–1.34]	4.06 × 10^−3^				56	1.23 [1.01–1.49]	4.34 × 10^−2^

* *n* = 64 M0 patients; ** the multivariate analysis was performed with FATE1 both as a binary variable (left) and as a continuous variable (right).

**Table 4 cancers-12-00689-t004:** Clinical, histopathological and serological features of adults with ACT in this study. For patients 29 to 34, serum samples labelled with *a* were taken before surgery, samples labelled with *b* after surgery.

Patient	Gender	Clinical Manifestations	Age at Diagnosis (Years)	Histopathology	ENSAT Stage	Surgical Resection	M	Recurrence	Outcome	DFS (Months)	FATE1 Abs IF+	FATE1 Abs ELISA+	FATE1 Abs WB+	FATE1 H-score
1	F	Unkn	51	ACC	3	Yes	Yes	Yes	DD	3	-	+	+	NA
2	M	C, Andro	18	ACC	3	Yes	Yes	Yes	DD	9	-	-	+	NA
3	F	C, Andro	45	ACC	2	Yes	Yes	Yes	Alive	96	+	+	+	NA
4	M	Andro, Estro	69	ACC	2	Yes	No	No	DOC	41	-	+	+	NA
5	F	C, Aldo	46	ACC	2	Yes	Yes	Yes	DD	24	+	+	+	NA
6	F	C, Andro	42	ACC	2	Yes	Yes	No	Alive	104	-	-	+	NA
7	M	Unkn	42	ACC	2	Yes	Yes	Yes	DD	16	+	+	+	NA
8	M	Unkn	52	ACC	3	Yes	Yes	No	Alive	98	+	+	+	NA
9	F	C, Andro	58	ACC	2	Yes	Yes	Yes	DD	8	-	-	+	NA
10	F	No	38	ACC	1	Yes	No	No	Alive	36	-	-	+	NA
11	F	No	47	ACC	3	Yes	Yes	No	Alive	30	-	-	+	NA
12	F	C	49	ACC	2	Yes	Yes	Yes	Alive	36	-	+	+	NA
13	M	Andro	60	ACC	2	Yes	No	Yes	DD	20	+	+	+	NA
14	F	C, Andro	44	ACC	1	Yes	Yes	Yes	Alive	69	+	+	+	NA
15	F	C	40	ACC	2	Yes	Yes	Yes	DD	11	-	-	+	NA
16	M	No	62	ACC	3	Yes	Yes	No	Alive	27	-	+	+	NA
17	F	No	77	ACC	2	Yes	Yes	No	Alive	92	+	+	+	NA
18	F	Unkn	70	ACC	2	Yes	Yes	No	Alive	80	+	-	+	NA
19	F	C, Andro	68	ACC	1	Yes	No	Yes	Alive	98	-	+	+	NA
20	F	Unkn	46	ACC	3	Yes	Yes	Yes	DD	3	-	-	+	NA
21	F	Unkn	42	ACC	3	Yes	Yes	Yes	Alive	29	-	+	+	NA
22	F	Unkn	78	ACC	3	Yes	Yes	Yes	DD	3	-	+	+	NA
23	F	Unkn	52	ACC	2	Yes	Yes	No	Alive	65	-	+	+	NA
24	F	No	70	ACC	3	Yes	Yes	No	Alive	62	-	-	+	NA
25	F	Unkn	53	ACC	2	Yes	Yes	No	Alive	35	-	+	+	NA
26	F	No	49	ACC	2	Yes	Yes	Yes	Alive	20	-	-	+	NA
27	F	No	30	ACC	2	Yes	Yes	No	Alive	4	+	+	+	NA
28	M	C, Estro	21	ACC	2	Yes	Yes	No	Alive	66	-	-	+	NA
29a	M	C	64	ACC	2	Yes	Yes	Yes	DD	48	+	+	+	0.3
29b											+	+	+	
30a	M	No	49	ACC	1	Yes	No	No	Alive	78	-	+	+	NA
30b											-	+	+	
31a	M	C	47	ACC	3	Yes	Yes	No	Alive	70	-	+	+	1.5
31b											-	+	+	
32a	F	Aldo	51	ACC	3	Yes	Yes	No	Alive	58	-	+	+	0
32b											-	+	+	
33a	M	C	61	ACC	1	Yes	No	No	Alive	86	-	+	+	3
33b											+	+	+	
34a	F	C, Andro	38	ACC	2	Yes	No	No	Alive	74	+	+	+	1.5
34b											-	+	+	
35	M	C	41	ACC	4	No	Yes	Yes	DD	5	+	-	+	NA
36	F	No	39	ACC	4	No	Yes	Yes	Alive	9	-	-	+	NA
37	F	C	60	ACC	4	No	Yes	Yes	DD	0	-	-	+	NA
38	F	C	57	ACC	4	No	Yes	Yes	DD	42	-	+	+	NA
39	M	No	79	ACC	4	No	Yes	Yes	Alive	0	-	-	+	NA
40	F	No	59	ACC	2	Yes	Yes	No	Alive	19	+	+	+	NA
41	F	No	60	ACC	4	No	Yes	Yes	DD	0	+	+	+	NA
42	F	C	43	ACC	4	No	Yes	Yes	DD	0	-	-	+	NA
43	M	No	53	ACC	4	No	Yes	Yes	Alive	0	-	+	+	NA
44	F	C	47	ACC	4	No	Yes	Yes	DD	0	-	+	+	NA
45	F	C	36	ACC	4	No	Yes	Yes	DD	0	+	+	+	NA
46	F	No	59	ACC	2	Yes	Yes	No	Alive	68	-	+	+	NA
47	M	No	58	ACA	NA	No	NA	NA	Alive	NA	+	+	+	NA
48	M	Aldo	67	ACA	NA	No	NA	NA	Alive	NA	+	+	+	NA
49	F	C	46	ACA	NA	No	NA	NA	Alive	NA	-	-	+	NA
50	F	C	35	ACA	NA	No	NA	NA	Alive	NA	-	+	+	NA
51	F	No	70	ACA	NA	No	NA	NA	Alive	NA	-	+	+	NA
52	M	No	32	ACA	NA	No	NA	NA	Alive	NA	-	+	+	NA
53	F	C	48	ACA	NA	No	NA	NA	Alive	NA	-	+	+	NA
54	F	C	34	ACA	NA	No	NA	NA	Alive	NA	+	-	+	NA
55	F	C	58	ACA	NA	No	NA	NA	Alive	NA	-	+	+	NA
56	M	No	54	ACA	NA	No	NA	NA	Alive	NA	-	+	+	NA

Clinical manifestations: Unkn, unknown; C, cortisol-secreting; Andro, androgen-secreting; Estro, estrogen-secreting; Aldo, aldosterone-secreting. Histopathology: ACC, adrenocortical carcinoma; ACA, adrenocortical adenoma. Outcome: DD, dead of disease. DFS: disease-free survival. NA: not available.

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
