# Peer review of "Cancer-testis Antigen FATE1 Expression in Adrenocortical Tumors Is Associated with A Pervasive Autoimmune Response and Is A Marker of Malignancy in Adult, but Not Children, ACC"

_cancers, 2020, doi:10.3390/cancers12030689_

Round 1
Reviewer 1 Report
Please check the attachment.

Author Response
We thank this reviewer for taking the time to read our manuscript, but we believe s/he missed the main points of our work. In synthesis, our findings allow to conclude that circulating antibodies against the FATE1 protein, a cancer-testis antigen, are present in all patients affected with adrenocortical tumours, both pediatric and adults, when investigated using the most sensitive method available (Western blot). We believe this is a relevant finding because of the cancer-restricted expression of this protein, which may be a novel target for immunotherapy in malignant adrenocortical tumours. In parallel, by analysis of publicly available gene expression datasets from patients with adrenocortical carcinoma (ACC), we confirmed and extended previous data showing that FATE1 overexpression is significantly associated with poor prognosis and identified transcripts that are significantly associated with FATE1 high and low expression, respectively. Remarkably, the first group is enriched with transcripts encoding proteins involved in steroidogenesis and the second group with immune response-associated transcripts.
1) Consistent with our most important findings, we truly believe that the manuscript’s title conveys the main message of our study.
2) Since the presence of circulating antibodies against FATE1 is widespread in patients with adrenocortical tumors and there is no prognostic significance, our results show that a pervasive immune response against this protein is present in those patients. We are not proposing to use the assay of anti-FATE1 antibodies as a clinical test to assess malignancy. Further studies (outside this scope of this paper) are required to assess the value of circulating anti-FATE1 antibodies as a diagnostic tool for adrenocortical tumors.
3) The question of specificity of a humoral anti-FATE1 immune response in patients with adrenocortical tumors compared to normal subject is interesting. However, it would be necessary to screen dozens of normal subjects to get at least a partial view of presence or not of anti-FATE1 antibodies in the normal population. This was clearly beyond the scope of this paper. In addition, we would like to underline here that in previous studies describing the presence of circulating anti-FATE1 antibodies in hepatocarcinoma patients (refs. 11 and 12 in our paper) only cancer patients, and not normal subjects, were investigated.
4) As shown in Table 1 in our paper, all childhood adrenocortical tumors where samples for IHC were available were positive for circulating anti-FATE1 antibodies using Western blot. Similarly for adult tumors (Table 4).
5) We never affirmed that the presence of circulating anti-FATE1 antibodies is a marker of aggressivce ACC, bur FATE1 mRNA expression in the tumor is, at least for adult cases (see Figure 3).
6) The methods used to analyze the correlation of FATE1 mRNA expression in ACC with clinical parameters are described in detail at page 17 in our manuscript, section 4.8. These methods are basically the same used in our previous published paper cited as ref. 21 in the present manuscript.
7) See point 5).
8) Frankly we don’t understand how it is possible to criticize the text at lines 241-264 in our manuscript, which is based on discussion of objective findings. So we decided to maintain that text in the revised version.
Reviewer 2 Report
Dear author,
I have read the article with high interest. The concept of the manuscript is novel and the experiments are well carried out. Hence I am recommending to accept it after minor revision.
Fig 5.a please add scale bar. Fig 5. c please quantify the WB expressions.
Author Response
We sincerely thank this reviewer for appreciating the novelty and the importance of our work and its potential impact in the field.
As suggested by this reviewer, we have added scale bars to the immunofluorescence images shown in Figure 5A. On the other hand, we believe it would not be of interest to quantify band signals in the Western blot shown in Fig. 5D. This is a qualitative, and not quantitative, representative experiment, which is only intended to show that sera from patients with adrenocortical tumors display an immunoreactive band against the FATE1 recombinant protein.
Reviewer 3 Report
This observational study investigated the cancer-testis antigen FATE1 in adrenocortical carcinoma patients in tumor cells and anti-FATE1 antibodies in systemic circulation, and their association with prognostic parameters. Despite the relatively low number of patients analyzed de novo (27 pediatric, 5 adults, plus 56 ACT patients for anti-FATE1 determination), findings presented here are novel and interesting also due to the inclusion of wider TGCA datasets.
However, few points should be addressed by authors:
- To detect circulating FATE1 in sera, authors used three different methodologies with somehow completely different results in terms of percentage of positivity and identification of positive patients. Could the authors discuss and address these discrepancies, also in view of a possible use of circulating FATE1 as surrogate biomarker? Which one of those methods actually have the most-powerful sensitivity and specificity? How many patients actually developed anti-FATE1 autoantibodies? This is a crucial point that needs to be properly addressed in order to confirm the authors claim that FATE1 expression is associated with a pervasive immune response
- In Materials and Methods, in Figure 5C, what numbers (1-15) above the WB lanes indicate? A better description of the Figure should be provided.
Author Response
We thank this reviewer very much for her/his positive evaluation of the novelty and importance of our study.
We described extensively in the text (at lines 108-114 for children – Table 1;, lines 178-184 for adult – Table 4) the presence of circulating anti-FATE1 antibodies in patients with adrenocortical tumors using each one of three assay methods (IF, ELISA and WB). No correlation existed between the presence of anti-FATE1 antibodies and clinical parameters. Further studies, clearly outside this scope of this paper, are required to assess the value of those antibodies as a diagnostic tool for adrenocortical tumors.
We have specified in Figure 5C legend that serum samples 1-15 were derived from patients 1-15 in Table 4.
Round 2
Reviewer 1 Report
In the revised manuscript, Doghman-Bougurerra and colleagues have not responded to the majority of this reviewers concerns or that of the other reviewers. The data in the manuscript reviews their attempts to correlate the transcript expression levels of the FATE1, a downstream target of SF1 and circulating antibodies to the protein as markers of aggressiveness in adrenocortical cancer.
- The organization of the data (if the current title is to be used) should be reversed. The authors could show the transcript changes of FATE1, high vs low predict disease aggressiveness in adult but not pediatric tumors (they should definitely give us their thoughts on why). Then show the differential signature of downstream targets suggesting an altered immune signature with high FATE1. They conclude FATE1 expression is associated with a “pervasive specific immune response” based upon analysis of associated changes in transcript levels of genes in FATE1 high vs FATE1 low samples. However, the weakness persists that no confirmation of changes of these genes at RNA or protein level are provided. It is not unexpected that a downstream target of SF1 would be associated with a hormonal more active and less active immune signature, but is unclear what the data as presented provide to the field for clinical care or understanding mechanism of disease.
- Then they could argue they decided to see if circulating antibodies to the protein might reflect disease activity. They attempted to detect FATE1 antibodies in the circulation by 3 methods (immunofluorescence, ELISA and Western blot), and only the western was consistent, but never quantitated. Importantly antibody was detected in benign adrenal adenomas and NO normal adrenal controls were assessed. The authors need to acknowledge these intensive efforts were a disappointment at this point. In addition, no data were provided to confirm that FATE1 by IHC expression correlates with FATE1 antibodies in human sera.
- The conclusions of the interpretation of the data across the manuscript continue to be overstated. The authors are to be commended for the extensive evaluation of FATE1 transcript, mRNA, protein and antibody changes as well as altered associated downstream transcript profiling, but the story is unfinished and not convincing at present of an important mechanistic or clinically relevant marker in ACC.
Author Response
We believe a major misunderstanding of this reviewer exists. In our paper, we showed that a specific anti-FATE1 immune response is detectable in patients with adrenocortical tumors using the most sensitive method for detection of circulating autoantibodies (Western blot). When we wrote about this specific anti-FATE1 immune response we did not by any means allude to the enriched presence of genes involved in immune response in the cluster of transcripts associated to high FATE1 mRNA expression in publicly available databases of ACC. These results are important since they confirm and extend on a large number (considering the rarity of the tumor) of patients with ACC the results published in our 2016 EMBO Rep. paper (cited as ref. 5 in the present manuscript), which showed a prognostic value of FATE1 overexpression (as measured by IHC) in adult ACC. On the other hand, hundreds of papers have been published exploiting cancer gene expression data present in public databases like the TCGA, similarly to our study, without providing confirmation at the single mRNA or protein level. This is clearly out of the scope of those and of our own study.
In any case, we thought we had clearly expressed those notions in the following passages of our manuscript :
- Abstract, lines 43-46
- page 2, lines 77-78
- page 8, lines 113 and following
- page 14, line 184 and following
- Sections 2.2 and 2.3
- page 15, lines 243 and following
However, with the purpose to wipe out any possible ambiguity, we changed the manuscript’s title into: «Cancer-testis antigen FATE1 expression in adrenocortical tumors is associated with a pervasive autoimmune response and is a marker of malignancy in adult, but not children, ACC ».
In our opinion, our manuscript is very clearly and logically structured, since it first reports results concerning FATE1 transcript/protein expression and presence of circulating autoantibodies in pediatric ACC (Section 2.1), then it moves to show results in adult ACC (Section 2.2) and finally reports the results of analyses in publicly available databases (Section 2.3). We then maintained its structure in the revised version.
Concerning the other points raised by this reviewer :
- we discussed the differences in FATE1 value as a prognostic marker in children vs . adult ACC at page 15, lines 247 and following;
- in our study we have shown that the presence of anti-FATE1 autoantibodies in patients with ACC is not correlated with disease stage or activity but is a widespread finding. Following this reviewer’s suggestion, we have exposed this notion more explicitely at page 8, lines 119-120 for pediatric tumors and at page 14, lines 190-191 for adult tumors;
- we have quantitated band intensities in Western blots shown in Fig. 5C and D;
- as we wrote in our previous rebuttal to this reviewer’s criticisms, the question of specificity of a humoral anti-FATE1 immune response in patients with adrenocortical tumors compared to normal subjects is interesting. However, it would be necessary to screen dozens of normal subjects to get at least a partial view of presence or not of anti-FATE1 antibodies in the normal population. This was clearly beyond the scope of this paper. In addition, we would like to underline here that in previous studies describing the presence of circulating anti-FATE1 antibodies in hepatocarcinoma patients (refs. 11 and 12 in our paper) only cancer patients, and not normal subjects, were investigated ;
- we don’t understand how this reviewer can write about a potential «disappointment» of ours since in this paper we present a large amount of work aimed at evidencing the presence of circulating anti-FATE1 antibodies using three different methods (IF, ELISA and Western blot). By these in-depth investigations we reached the important conclusion that a pervasive immune directed specifically against the FATE1 cancer-testis antigen is present in patients with adrenocortical tumors. We are then rather satisfied and not disappointed about our results;
- the data about presence of circulating anti-FATE1 antibodies and FATE1 protein expression, as detected by IHC, are shown in Table 1 (pediatric ACC) and Table 4 (adult ACC);
- we thank this reviewer for her/his appreciation of our work. However, we profoundly disagree with her/him since we believe that our study is innovative and important since on one side it shows the widespread presence of circulating anti-FATE1 antibodies in patients with ACC and on the other side it shows that tumor FATE1 mRNA expression is a robust marker of maligancy and its expression levels are linked to distinct sets of coregulated transcripts. Given its immunogenic properties, FATE1 stands out as a potential tumor-specific target for immunotherapy in ACC. As to the mechanistic role of FATE1 in ACC malignancy, it has been dissected in our previously published EMBO Rep. paper (cited as ref. 5 in the present manuscript).
Round 3
Reviewer 1 Report
N/A